# Bio-Compatibility and Bio-Insulation of Implantable Electrode Prosthesis Ameliorated by A-174 Silane Primed Parylene-C Deposited Embedment

**DOI:** 10.3390/mi11121064

**Published:** 2020-11-30

**Authors:** Chin-Yu Lin, Wan-Shiun Lou, Jyh-Chern Chen, Kuo-Yao Weng, Ming-Cheng Shih, Ya-Wen Hung, Zhu-Yin Chen, Mei-Chih Wang

**Affiliations:** 1Institute of New Drug Development, China Medical University, No.91 Hsueh-Shih Road, Taichung 40402, Taiwan; janet_emily_1@yahoo.com.tw; 2Master Program for Biomedical Engineering, Collage of Biomedical Engineering, China Medical University, No.91 Hsueh-Shih Road, Taichung 40402, Taiwan; 3Biomedical Technology & Device Research Laboratories, Industrial Technology Research Institute, No.195, Sec.4, Chung Hsing Rd., Chutung, Hsinchu 31057, Taiwan; WanShiunLou@itri.org.tw (W.-S.L.); hugoshiu@itri.org.tw (M.-C.S.); ya-wen@itri.org.tw (Y.-W.H.); 4ACE Biotek Co., Ltd., No.66, Shengyi 2nd Rd., Jhubei, Hsinchu 30261, Taiwan; jc_chen@acebiotek.com; 5Technology Transfer and Law Center, Industrial Technology Research Institute, No.195, Sec.4, Chung Hsing Rd., Chutung, Hsinchu 31057, Taiwan; allan_weng@itri.org.tw

**Keywords:** interdigitated electrode, retinal prosthesis, parylene-C, bio-compatibility, bio-insulation

## Abstract

Microelectrodes for pain management, neural prosthesis or assistances have a huge medical demand, such as the application of pain management chip or retinal prosthesis addressed on age-related macular degeneration (AMD) and the retinitis pigmentosa (RP). Due to lifelong implanted in human body and direct adhesion of neural tissues, the electrodes and associated insulation materials should possess an ideal bio-compatibility, including non-cytotoxicity and no safety concern elicited by immune responses. Our goal intended to develop retinal prosthesis, an electrical circuit chip used for assisting neural electrons transmission on retina and ameliorating the retinal disability. Therefore, based on the ISO 10993 guidance for implantable medical devices, the electrode prosthesis with insulation material has to conduct bio-compatibility assessment including cytotoxicity, hemolysis, (skin) irritation and pathological implantation examinations. In this study, we manufactured inter-digitated electrode (IDE) chips mimic the electrode prosthesis through photolithography. The titanium and platinum composites were deposited onto a silicon wafer to prepare an electric circuit to mimic the electrode used in retinal prosthesis manufacture, which further be encapsulated to examine the bio-compatibility in compliance with ISO 10993 and ASTM guidance specifically for implantable medical devices. Parylene-C, polyimide and silicon carbide were selected as materials for electrode encapsulation in comparison. Our data revealed parylene-C coating showed a significant excellence on bio-insulation and bio-compatibility specifically addressed on implantable neuron stimulatory devices and provided an economic procedure to package the electrode prosthesis. Therefore, parylene C encapsulation should serve as a consideration for future application on retinal prosthesis manufacture and examination.

## 1. Introduction

The implantable medical electronics clinically are mostly related to the neuron stimulatory devices, such as electronic ear, retinal prosthesis, pain-controlled electronic devices and implantable heart de-vibration devices. Overall business market size is expected to reach 10.5 billion by 2026, and rising at a market growth of 12.5% compound annual growth rate (CAGR) during the forecast period [1]. The implantable neuron stimulatory devices are still existing many unmet medical needs, such as insufficient bio-insulation, bio-compatibility, anti-inflammatory capability and the optimal implantation-assisted hand-helds devices.

Addressed on bio-compatibility as well as bio-insulation of electrode prosthesis to resist the body fluid invasion during the long-termed grafted in vivo, the encapsulation materials directly contact with tissues surrounding the implanted site. Therefore, whether the prosthesis interface possesses an ideal bio-compatibility with low cytotoxicity and safety concerns, meanwhile exhibit excellent waterproof capability emerged as an important consideration. One of the encapsulation materials, polyimide, having been applied on the medical device coating, showed good bio-compatibility, easily manufacturing and stable for materials used in typical microelectro-mechanical system (MEMS) [2]. However, there are some problems associated with the flexible polyimide coating, such as stiffness, durability and long-termed insulation capability in vivo, and lead to need further material modification [3,4]. Another material, silicon carbide encapsulation has been widely used in the semi-conductor assembly process, and showed excellent insulation capability as described [5], which is emerging as an enabling material for biomedical microsystems due to its unique combination of electrical, mechanical and chemical properties. Surface modification with silicon carbide showed its bio-compatibility, which did not undergo consistent oxidative stress events and did not exhibit morphological modifications or adverse reactions in mitochondrial membrane potential [6]. 

Parylene-C has emerged as a promising material to encapsulate the implantable devices due to its unique mechanical properties and inertness [7], excellent insulation capability to resist the body fluid invasion and bio-compatibility. Parylene-C possesses the advantages of providing an entirely conformal, durable, pinhole-free material for coating of extreme utility with an exceptional range of materials, products and purposes. Despite its many advantages, parylene-C’s chemical structure leads to the inferior adhesion to many biomaterials and shows non-reliable interface adhesion required for optimal performance. Implementing optimal adhesion of parylene-C to biomaterials or medical devices requires surface modification of devices or application of adhesion promoting agents. The chemical monolayer silane A-174 (3-methacryloxypropyltrimethoxysilane—C_10_H_20_O_5_Si) is used most frequently serving as an adhesion promoting agent to modify the device surfaces and improve parylene-C adhesion [8].

Microelectrodes firstly are classified into several types based on the constructed shape, including spherical, hemispherical, line, disc, and ring types, with distinct application purpose and different micro-molecules diffusion, transmission patterns. Moreover, one of the important design considerations is the micropatterning of electrodes arrangement, which can be classified into individually addressable arrays and integrated arrays. The microelectrodes deployment on the array could be further divided into ordered and random array, respectively. The ordered microelectrode array has regular interval space between electrodes with advantages of highly reproducible, reliable, cheap and easy to be manufactured by silicon technology in a large scaled production, such as photolithography [9]. Ordered microelectrode array is often used in the biological studies, theoretical simulation and electrochemistry analysis.

According to ISO 10993 guidance for the development of medical devices, bio-compatibility examinations and safety concerns evaluation are mandatory to conduct before the devices proceed the clinical trials [10]. Our study intended to develop retinal prosthesis, an electrical circuit chip used for assisting neural electrons transmission on retina and ameliorating the retinal disability. Therefore, based on the classification of implantable medical devices, the developed retinal prosthesis has to conduct bio-compatibility assessment including cytotoxicity, hemolysis, irritation and pathological implantation examinations [11,12,13,14,15].

In this study, we manufactured inter-digitated electrode (IDE) chips mimic the electrode prosthesis through photolithography to deposit titanium and platinum composite onto a silicon wafer to prepare an electric circuit to mimic the electrode used in retinal prosthesis manufacture, which further be encapsulated to examine the bio-compatibility in compliance with ISO 10993 and ASTM guidance specifically for implantable medical devices. Parylene-C, polyimide and silicon carbide were selected as materials for electrode encapsulation in comparison.

## 2. Materials and Methods

### 2.1. Manufacture Electrode Prosthesis

Photolithography is currently the most prevalent method used to manufacture microelectrode array, which deploys electrode materials such as gold, platinum, iridium or carbon on silicon matrix through vapor deposition or sputtering techniques, and etches the electrode in a particular pattern through the protection of photoresist. Finally, the electrode matrix is bathed in acid or exposed to oxygen plasma to remove the photoresist, and packaged with Si_3_N_4_ or SiO_2_ by plasma enhanced chemical vapor deposition (PECVD) for insulation [16].

The fabrication process of interdigitated microelectrode array used in this study is outlined in Figure 1A, manufactured using 4-inch silicon wafer with a 1 μm silicon oxide epitaxial layer as basal matrix. The electrode layout was designed by Virtuoso Layout Suite (Cadence Design Systems, Brack- nell, UK) to manufacture 10 mm × 30 mm chips featuring band and gap widths in 200 μm. The electrode was patterned in a conventional mask aligner as previous described with slight modification [17] and processed in a few more post-processing steps to deposit parylene-C (Appendix Aa) on interdigitated electrodes (IDE) chip to make the potentially biocompatiable. An adhesion promoter, γ-methacryloxypropyltrimethoxy (Appendix Ab) (A-174 silane) (Momentive Performance Materials Inc., Waterford, NY, USA) is used to improve the adhesion of parylene to the silicon oxide passivation layer. Parylene-C is selected due to its mechanical flexibility (Young’s modulus ∼3 GPa). Briefly, the silicon basal matrix was deposited with a 5–7 μm thick parylene-C for insulation and coated with a 0.5–2 μm thick SU-8 photoresist layer. Subsequently, the structures were defined lithographically in a Nikon NSR-2205i12D stepper, then a 100 nm of platinum were deposited over 10 nm of titanium layer by e-beam evaporation over the entire basal matrix. The metal lift-off was carried out by immersion of the basal matrix in a stripper solution bath that left the metallic electrode structures on the wafer (Figure 1B). In order to remove traces of SU-8 photoresist layer after development, wafers were de-scummed in a mild oxygen plasma before dicing them into final size. After dicing, the IDE chips were again insulated with parylene-C and A-174 silane for electrode protection (Figure 1C). All electrodes were cleaned by immersion in 70% ethanol and rinsed several times in sterile deionized (DI) water, followed by sterilization with ethylene oxide (EtO), and outgassing for at least 48 h prior to implantation. The conformal parylene-C, polyimide and silicon carbide coating were conducted in compliance with the standard protocol developed by manufacturer (PDS 2010 Conformal Coater, Cookson Electronics Inc., Jersey, NJ, USA) and previous publish with slight modification [2,6,18].

### 2.2. Saline Soaking Test

To examine the durability and water-proofing capability of IDE chip in the physiological scenario, an accelerated test was conducted to immerse the IDE chip at 70 °C and 90 °C respectively in saline for 45 days as illustrated in Figure 2A. Top was filled with mineral oil to prevent saline vaporization. The insulation capability resisting to water invasion was first examined in gross picture and then examined the direct current resistance (DC-R) in a pre-voltage and DC auto-loaded mode in compliance with the instruction of instrument (34401A, Agilent Inc., Santa Clara, CA, USA). 

### 2.3. Cytotoxicity Evaluation of Bio-Compatibility Test

Detailed information is provided in the Appendix A. 

### 2.4. Hemolysis Evaluation of Bio-Compatibility Test 

Detailed information is provided in the Appendix A.

### 2.5. Irritation Evaluation of Bio-Compatibility Test

According to ISO10993-10 and ASTM F749-98 guidance, implantable medical device should be subjected to irritation assessment which evaluate the erythema (ER) and edema (ED) raised from devices implantation. Again, due to the high sensitivity, rabbit model is considered as the first choice for irritation test of medical devices. The encapsulated IDE chip was immersed in 3 mL normal saline and sesame oil, respectively for 72 h at 37 °C to prepare the eluted concentrates, and the extraction solution was served as control group. Rabbit was anesthetized with 2% isoflurane (Abbott, Chicago, IL, USA), back hair was shaved, 0.2 mL extracted or controlled solution was injected separately into the back subcutaneous skin in 5 sites with 2 cm interval distance. The gross appearance of rabbit skin was recorded and evaluated at 0, 24, 48 and 72 h post-injection. The ER and ED responses were evaluated according to criteria shown in Results to determine the score, and mean of ED plus ER score was calculated for comparison of irritation, difference between test and control ≤1 could be recognized as no irritation. All animal experiments were approved by the Industrial Technology Research Institute (ITRI) Committee for the Use and Care of Animals.

### 2.6. Bio-insulated Evaluation of Implantation Test 

To examine the bio-insulation capability of IDE chip encapsulated by parylene-C in vivo, the IDE chips were implanted subcutaneously in rabbit back skin, and collected for direct current resistance measurement and gross appearance observation at 12 weeks post-implantation (wpi). Rabbit was anesthetized with 2% isoflurane for surgery, and received analgesic drug (buprenorphine, 0.1 mg/kg/day for 2 days) after operation. The insulation capability resisting to body-fluid invasion was examined by direct current resistance (DC-R) in a pre-voltage and DC auto-loaded mode in compliance with the instruction of instrument (34401A, Agilent, Inc., Santa Clara, CA, USA).

### 2.7. Hypersensitivity Evaluation of Implantation Test 

According to ISO 10993-6 and ASTM F981-04 guidance for hypersensitivity evaluation of implantable medical device, the IDE chips were implanted subcutaneously in rabbit back in procedure similar to Section 2.6. Four IDE chips were implanted in the dorsal subcutaneous tissue along right side of the spine, 25–50 mm from the midline and parallel to the spine, and about 25 mm from each other. The identical-sized HDPE implantation was served as negative controlled group, implanted along the left side of the spine. All animals used in this study were observed daily and sacrificed at 12 weeks after implantation to evaluate the absorption and local effects by microscopic observation of H&E staining (Nikon E200, Nikon, Tokyo, Japan). Histopathological assessment was conducted by three individual pathologists to grade the invaded cell types and responses, and neovascularization, fibrosis and fat infiltration, the score was used to calculate the total sum to represent the overall irritant scores, which ≤2.9 could be considered as non-irritant. 

### 2.8. Statistical Analysis

Data are presented as mean ± SD, statistical comparisons were performed by Student’s t-test or one-way analysis of variance (ANOVA) and p values <0.05 were considered significant. All calculations were performed using Statistics Analysis System (SAS) licensed to China Medical University. All in vivo data are representative of at least 3 independent experiments as indicated.

## 3. Results

### 3.1. Electrode Prosthesis with Parylene-C Coating and Accelerated Fatigue Examination

The inter-digitated microelectrode array was manufactured from 4-inch silicon wafer, as illustrated in Figure 1, and the structure of parylene-C and A-174 were also illustrated (Appendix A). The IDE chips embedded with parylene C were observed by SEM, showed approximately 7.3 μm in thickness and uniform surface (Appendix A). To examine the liability in the accelerated scenario, embedded chips were soaked in saline and top filled with mineral oil to prevent vaporization (Figure 2A). Before saline soaking, the gross chips surface was observed, showing chips integrity, shining surface and luster (Figure 2B). The saline immersed chips were subsequently stored in 70 °C and 90 °C oven, respectively, for continuing 45 days. Data revealed the SiC and polyimide embedded chips showed apparent surface coating break, but parylene C coating group showed complete surface integrity (Figure 2C). We further examined the direct current resistance (DC-R) of chips collected from the saline soaking, showed diminished DC-R values in the SiC and polyimide embedded chips, but DC-R approximately = ∞ in the parylene C embedded group, and demonstrated the water-proof and electrode protection capability of embedment materials using parylene C were superior than that of the SiC and polyimide.

### 3.2. Cytotoxicity Examination of Electrode Prosthesis 

According to implantable medical device safety concern regulated by ISO 10993-5: 2009 (Biological evaluation of medical devices—Part 5: Tests for in vitro cytotoxicity. https://www.iso.org/standard/36406.html) and ASTM F813- 07(2012) (Standard Practice for Direct Contact Cell Culture Evaluation of Materials for Medical Devices. https://www.astm.org/DATABASE.CART/HISTORICAL/F813-07R12.htm), cytotoxicity resulted from the devices long-term grafted in vivo was the first issue to be explored. The L929 mouse fibroblast cells were seeded on 6-well plate, treated with eluted concentrates extracted from parylene C and polyimide embedded IDE chips, respectively. The cytotoxic morphology was observed at 24 h by crystal violet staining, and Latex rubber and HDPE tube extracted solutes were served as positive and negative comparison, respectively. Data revealed solutes extracted from parylene C and polyimide showed no morphological difference with the negative control (Figure 3A). The images were further analyzed and examined the cytotoxic grade by image J according to the qualitative morphological grading (Appendix A), demonstrating either parylene C or polyimide embedded IDE chips possesses grade 0 cytotoxicity (Table 1). Quantitative cell viability measurements by colorimetric assays also demonstrated cytotoxicity-free of parylene C or polyimide embedded IDE chips (Figure 3B,C).

### 3.3. Hemolysis Examination of Electrode Prosthesis 

According to ISO10993-4:2002 /AMD 1:2006 (Biological evaluation of medical devices—Part 4: Selection of tests for interactions with blood. https://www.iso.org/standard/39205.html) and ASTM F756-08 (Standard Practice for Assessment of Hemolytic Properties of Materials. https://www.astm.org/DATABASE.CART/HISTORICAL/F756-08.htm), all of the implantable medical devices intended to directly contact with blood tissue, mandatory to conduct the hemolytic assessment of the devices eluted concentrates. The electrode prosthesis in this study would like to mimic the retinal prosthesis which should be encapsulated sophisticatedly and implanted for at least 10 years orthotopically in vivo. Therefore, except of the cytotoxicity, whether the elutes from the embedded materials toxic and harmful to the blood cells have to be explored. In the hemolytic assessment, we added the eluted concentrates extracted from parylene C and polyimide embedded IDE chips to RBC solution, and subsequently measured hemoglobin released from the RBC disruption. Data revealed the blank corrected hemolysis of parylene C and polyimide near the blank and negative control groups, showing no statistical difference (*p* > 0.05) (Figure 4). The overall hemolysis % was subtracted negative control to calculate the hemolytic index, while parylene C and polyimide groups showed 0.09 and 0.31 in the hemolytic index, respectively (Table 2). Compared to the positive control showing extremely high reached 98.48 in the hemolytic index, parylene C and polyimide possessed non-hemolysis property (Appendix A). Due to the durability possessed by parylene C encapsulation, which was the only material examined further in the following detailed bio-compatibility assessments.

### 3.4. Irritation Examination of Electrode Prosthesis 

To examine the irritate properties of implantable medical device in compliance with ISO 10993-10:2010 (Biological evaluation of medical devices—Part 10: Tests for irritation and skin sensitization. https://www.iso.org/standard/40884.html) and ASTM F749-98 (2007) guidance (Standard Practice for Evaluating Material Extracts by Intracutaneous Injection in the Rabbit. https://www.astm.org/DATABASE.CART/HISTORICAL/F749-98R07E1.htm), the eluted concentrates extracted from the IDE chips were used for subcutaneous injection in rabbit. The sodium chloride (SC) and sesame oil (SO) extracted solutions were injected in five points in rabbit back skin right and left zone, respectively (Figure 5A). The skin appearance was evaluated accordingly (Appendix A) to determine and score the erythema and edema scenarios resulted from IDE elutes injection, and recorded the skin appearance at 0, 24, 48 and 72 h post-injection. All of the skin appearance showed no significant symptoms of erythema and edema in both of the SC and SO extractions (Figure 5B). Mean of scoring from the five injection points was used to evaluate the erythema and edema status, revealing means = 0 in the SC extracts in either erythema or edema evaluation (data no shown). Meanwhile, the SO extracts showed means = 0 in the edema evaluation, but mild erythema scoring (Figure 5C). Sum of mean value of erythema and edema scoring from IDE chips was used to compare with that of the controlled solution injection, and data showed difference of mean lower than 1, demonstrating no irritation symptom resulted from IDE chip extracts (Table 3).

### 3.5. Bio-Insulation Examination of Electrode Prosthesis through Subcutaneous Implantation 

To examine the insulation capability of parylene C embedded IDE chips in vivo, the IDEs chips were implanted subcutaneously in rabbit back in compliance with the ISO 10093-6 guidelines for implantable medical device. At 12 weeks post-implantation, the IDE chips were collected for direct current resistance (DC-R) measurement and surrounding tissue macroscopic assessment, which showed no any significant sign of inflammation, hemorrhage, necrosis, discoloration and clear interface between the chips and the surrounding tissues (Figure 6). Totally, five over eight IDE chips showed the electrical resistance = ∞, which revealed intact insulation and resist to the bio-fluids infiltration (Table 4). Three of eight IDE chips showed decreased electrical resistance, which might result from the retrieve surgery on rabbit back damaging the insulation layer while removing the transparent mucous membrane surrounding the IDE chips.

### 3.6. Hypersensitivity Examination of Electrode Prosthesis through Subcutaneous Implantation 

According to ISO 10993-6:2007 (Biological evaluation of medical devices—Part 6: Tests for local effects after implantation. https://www.iso.org/standard/44789.html) and ASTM F981-04 guidance for implantable medical devices bio-compatibility assessment (https://www.astm.org/DATABASE.CART/HISTORICAL/F981-04R10.htm), the parylene C embedded IDEs chips were implanted subcutaneously in rabbit dorsal tissues, compared with negative controlled PE tube for histopathological evaluation as illustrated (Figure 7A). The H&E staining showed no any significant signs of necrosis, immunocytes infiltration, infection and hemorrhage in both of the IDE chip and PE tube implantation (Figure 7B, Appendix A). Accordingly, the infiltrated immunocytes were characterized and scored (Appendix A), and the scenarios of neovascularization, fibrosis and fatty infiltrate were further examined and recorded from 0–4 (Appendix A) to determine the overall irritant status of implanted IDE chips (Table 5). Data revealed the histopathological evaluation at 12 weeks post-implantation, and the score of semi-quantitative evaluation system was 0.83 (Table 5), which inferred that the parylene C embedded IDE chips showed non-irritation in subcutaneous tissue (classified by Appendix A).

## 4. Discussion

Our study showed an affordable and facile engineering process to manufacture an IDE chip through photolithography (Figure 1) and subsequently insulated with parylene-C, polyimide or silicon carbide for waterproof comparison. Our data demonstrated the electrode prosthesis encapsulated with parylene-C possesses the superior protection capability in the accelerated aging experiment than that of encapsulated with polyimide and silicon carbide (Figure 2), which distinct from previous findings [19,20]. Since previous studies delineated the parylene-C has relatively higher water vapor transmission rate (WVTR) compared to many insulating materials, therefore it may not be sufficient to protect implanted ICs. Our data ameliorated the protection capability of parylene-C coating which may derived from the specific formulation, prime parylene-C adhesion and coating on silicon matrix. The coating increased the thickness of deposited protection layer (Appendix A), elongated the waterproof (Figure 2) and body fluid resistant period (Figure 6). Furthermore, the cytotoxicity (Figure 3), hemolysis (Figure 4), irritation (Figure 5) and implantation (Figure 7, Appendix A) assessments following ISO 10993 and ASTM guidance, electrode prosthesis insulated with parylene-C provide excellent bio-compatibility to mammalian cells, and resistance to vapor and body fluid invasion.

Microelectrodes for pain management, neural prosthesis or assistances have a huge medical demand, such as the application of retinal prosthesis and pain controllable chip [21,22,23]. These long-termed or lifelong implantable microelectrodes medical devices included the neuron recording or neuron stimulating electrodes. Due to lifelong implanted in human body, neural tissue initially recognized this implanted device based on the physiological and immune responses, and eventually encapsulated the device to result it dysfunction. From the pathological finding, the initial implantation surgery of electrode prosthesis resulted in surrounding tissue inflammatory responses. Consequently, this would stimulate the astrocyte and microglial cell from resting stage to reactive situation, and the electrode would subsequently be attached and encapsulated a glial cell layer approximately 10 μm in thickness. Meanwhile, neurons start to necrosis, resulting in magnitude neuron loss, and a glial cell- insulating layer would impede the diffusion of neural molecules and increase the electrical resistance, which eventually resulted in malfunction of electrode.

Two strategies could be considered to avoid the immune responses and resulted electrode malfunction, which were elicited from the implantation of electrode prosthesis. The first was optimal electrode material and shape construction. Due to the direct adhesion of neural tissues surrounding the implanted electrode and the associated encapsulation materials, the electrode material should possess an ideal bio-compatibility, including non-cytotoxicity and no safety concern elicited by immune responses. Besides, size, shape, cross-sectional area of electrode and texture of the electrode tip are significantly impact the immune responses. Second, coating modification of the implanted electrode prosthesis will improve the bio-compatibility. Optimal material modification is to conjugate anti-inflammatory drug, such as dexamethasone, which ameliorate the inflammatory responses elicited from prosthesis implanting surgery, or to conjugate particular extracellular matrix, such as neuronal adhesion molecule to prevent the attachment and growth of glial cells, and selectively to enhance the adhesion and growth of neurons. Moreover, the electrode prosthesis embedding materials modified with neuron committed growth factor or chemoattractant has been demonstrated to significantly promote the survival rate of neurons surrounding the implanted electrode prosthesis [24]. Therefore, to explore the most optimal bioactive molecule and drug delivery carrier used for the modification of embedding materials is the most important research field that material scientists addressed on.

Parylene-C has been widely used in medical devices coating to compart the inflammation eliciting components arose from medical devices in contacting with body fluids [18]. Parylene C coating was applied in manufacture of brain probe, bone screw, bone nail, guide pin, electrical circuits, and in promoting lubricity of medical device, such as syringe needle…etc. The adhesion molecule assists parylene-C coated onto many biomaterials and shows reliable interface adhesion required for optimal bio-insulation performance, played critical factor to determine the biostability and biocompatibility after implantation in vivo [25,26]. The A-174 silane is most frequently used as an adhesion promoting agent to modify the device surfaces and improve parylene-C adhesion on bio electronic devices. A feasibility study demonstrated parylene C was used to encapsulate a biomechanical energy harvesting medical device to serve as a long-term implantable energy nanogenerator in vivo [27]. Parylene-C has been coated on absorbable bone fixation devices manufactured by calcium phosphate fibers and showed cytotoxicity-free [28]. Another study was to modify parylene-C with fibronectin to improve the cell compatibility, while elevating the NIH-3T3 fibroblast and AML-12 hepatocyte adhesion, which demonstrated the property suitable for the development of cell-based microdevice [7]. Unsworth et al. used micropatterned Parylene-C/silicon dioxide composite substrate to support the growth and adhesion of hNT neurons, which could mimic the neural network and be served as an in vitro human brain pathological studying model. This demonstrated the non-cytotoxic characteristic of parylene-C and its potential application on medical device [29].

Collectively, we compared three types of popular bio-insulation substances used for electronic devices package and provided a distinct sight of parylene-C coating from previous publish [30]. Our manufacture process is easy to scaled-up, which is important for bioelectronic industry, as emphasized by another study [31]. We believe this development would offer a reference and a cue to the scientists addressed on the bioelectronics R&D and a hope to patients suffering neuron related diseases.

## 5. Conclusions

Our data provided that an economic procedure was established to package an electrode prosthesis, which would serve as highly potential for future application in retinal prosthesis manufacture and compliance with the ISO 10993 regulation for implantable medical devices. Parylene-C coating showed a significant excellence on bio-insulation and bio-compatibility specifically addressed on implantable neuron stimulatory devices.

## Figures and Tables

**Figure 1 micromachines-11-01064-f001:**
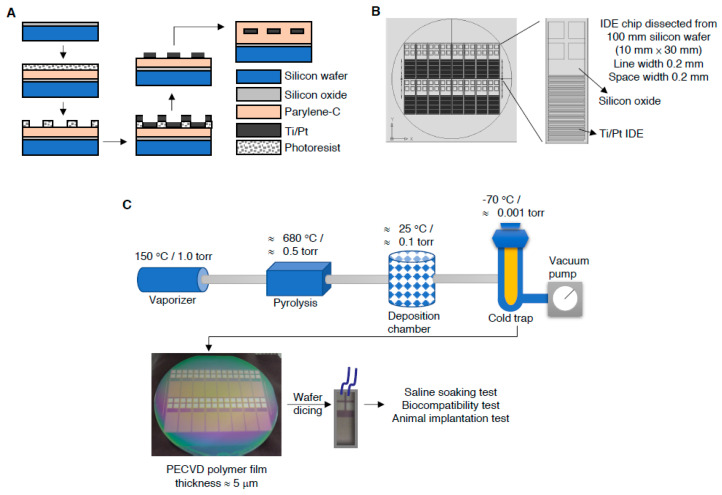
Manufacture process and encapsulation of IDE chips. (**A**) Schematic illustration of IDE chip design. (**B**) Metallic electrode structures of IDE chips were created on the 4-inches wafer by photolithography. (**C**) Parylene-C, polyimide, and silicon carbide were deposited on the IDE chips for insulation through chemical vapor deposition.

**Figure 2 micromachines-11-01064-f002:**
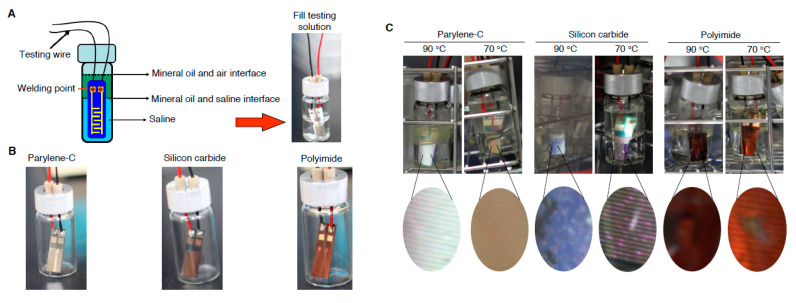
Accelerated aging experiment to examine the durability of IDE chip with parylene-C, polyimide and silicon carbide encapsulation. (**A**) Schematic illustration of experimental design to immerse the IDE chip in saline. (**B**) Representative gross appearance of IDE chips before immersed in saline. (**C**) Parylene-C, polyimide and silicon carbide insulated IDE chips were immersed in 70 °C and 90 °C saline for 45 days. (Magnification: 40×).

**Figure 3 micromachines-11-01064-f003:**
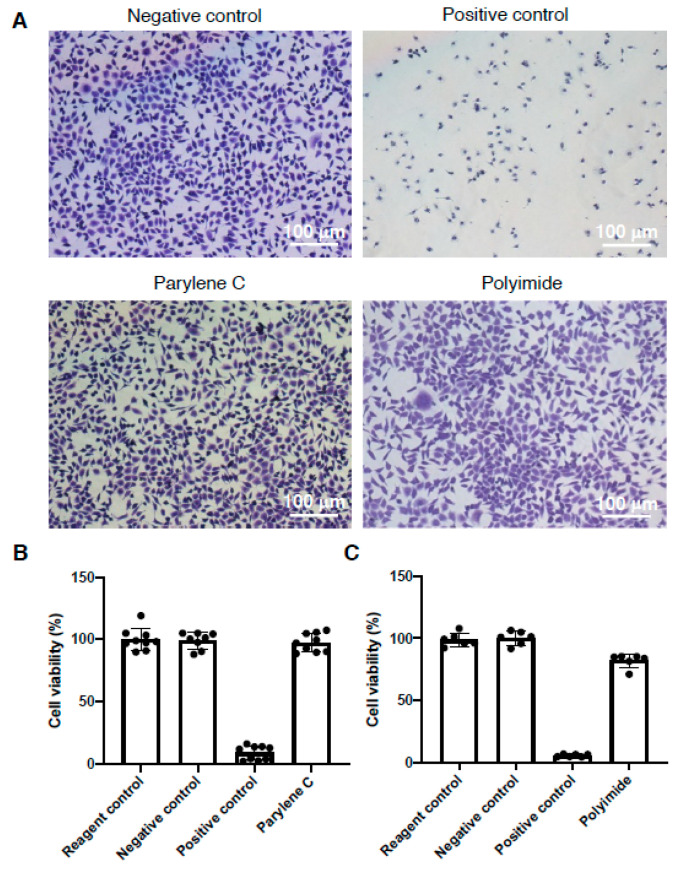
ISO 10993-5 cytotoxicity assessment using eluted concentrates of parylene-C and polyimide insulated IDE chips. (**A**) Representative images show crystal violet staining using L929 cells treated with eluted concentrates of parylene-C and polyimide insulated IDE chips for 24 h. (**B**) Colorimetric L929 cell viability assay using parylene-C eluted concentrates. (**C**) Colorimetric L929 cell viability assay using polyimide eluted concentrates. Data are presented as mean ± SD. (N > 6).

**Figure 4 micromachines-11-01064-f004:**
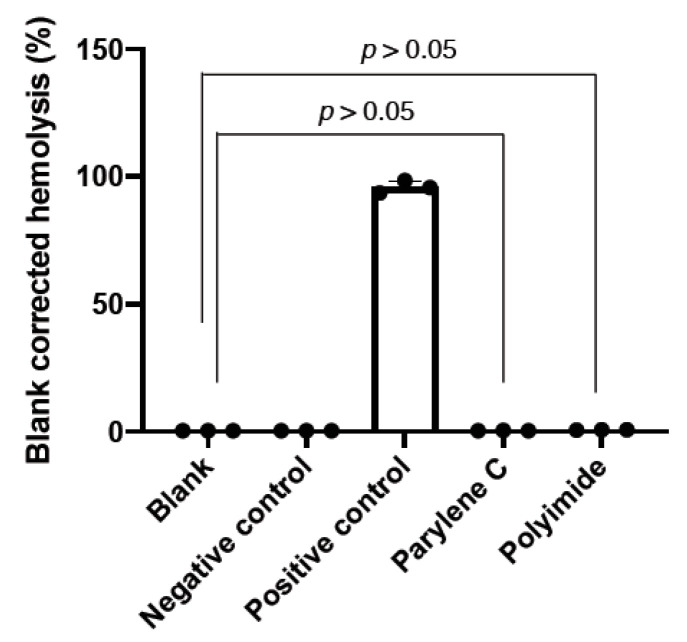
ISO 10993-4 hemolytic assessment using eluted concentrates of parylene-C and polyimide insulated IDE chips. The encapsulated IDE chips were immersed in 10 mL PBS for 72 h at 37 °C to prepare the eluted concentrates, which were mixed with 1 mL whole blood from rabbits for 3 h at 37 °C for hemolytic assessment. Data are presented as mean ± SD. (N = 3).

**Figure 5 micromachines-11-01064-f005:**
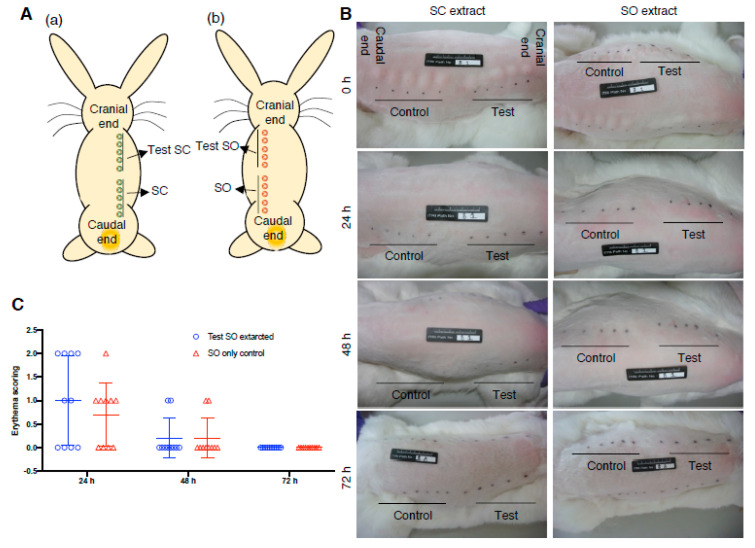
ISO 10993-10 irritation assessment using eluted concentrates of parylene-C insulated IDE chips. (**A**) Illustration of injection sites, extraction carrier only and eluted concentrates were injected in 5 sites, respectively. (**B**) Representative gross skin appearance at 0–72 h post-injection for evaluation of ED and ER. (**C**) ER score at 0–72 h post-injection, each circle or triangle represents the score evaluated from each individual injection site. SC: Saline as extraction carrier; SO: sesame oil as extraction carrier. ED: edema; ER: Erythema. Data are presented as mean ± SD. (N = 2).

**Figure 6 micromachines-11-01064-f006:**
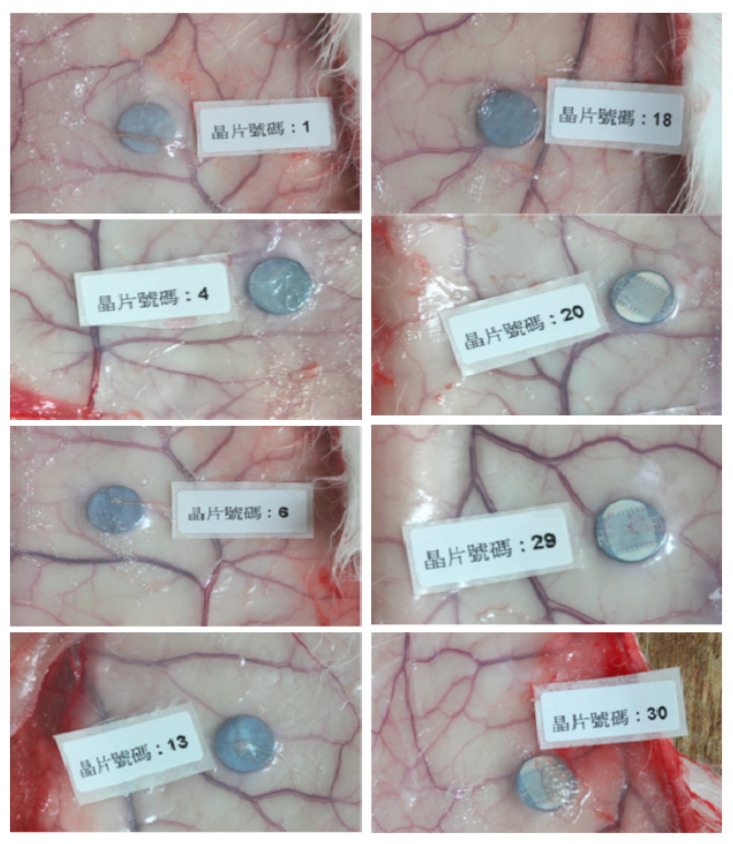
ISO 10993-6 implantation assessment to evaluate the bio-insulation capability. The insulated IDE chips were implanted subcutaneously in rabbit back, and retrieved at 12 weeks post-implantation for direct current resistance measurement and gross examination of surrounding tissues. (N = 8).

**Figure 7 micromachines-11-01064-f007:**
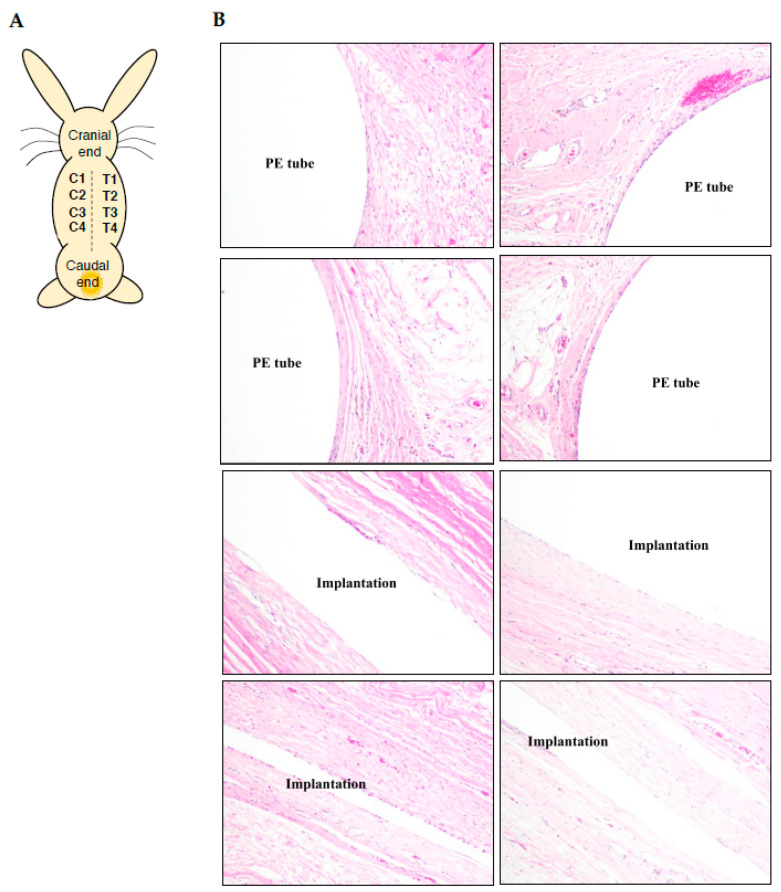
ISO 10993-6 Hypersensitivity examination of electrode prosthesis through subcutaneous implantation. (**A**) Illustration of implantation sites on rabbit back. T: insulated IDE chip; C: HDPE controlled sample. Number indicates implantation repeats. (**B**) H&E staining of subcutaneous tissues surrounding the graft at 12 weeks post-implantation. (N = 3). White area indicates the implantation site, “PE tube” indicates HDPE implantation, “Implantation” indicates IDE chip implantation. (Magnification: 100×).

**Table 1 micromachines-11-01064-t001:** Cytotoxic analysis of elutes extracted from polymer embedded chips.

Well	Percent Rounding	Percent Lysis	Grade	Reactivity
Negative control	0	0	0	None
Positive Control	100	100	4	Severe
Reagent Control	0	0	0	None
Parylene C-Silicon Wafer	0	0	0	None
Polyimide-Silicon Wafer	0	0	0	None

**Table 2 micromachines-11-01064-t002:** Result of hemolytic analysis.

Sample	Hemolytic Index
Blank	0.01
Negative Control	0
Positive Control	98.48
Parylene C-Silicon Wafer	0.09
Polyimide-Silicon Wafer	0.31

**Table 3 micromachines-11-01064-t003:** Irritant evaluation, C ≤ 1: no irritation.

Extracted Solution	Mean of Tested Material (ED + ER) (A)	Mean of Controlled Solution (ED + ER) (B)	Difference of Mean (Tested-Controlled, A-B) (C)
Normal saline	0.00	0.00	0.00
Sesame oil	0.2	0.15	0.05

**Table 4 micromachines-11-01064-t004:** Direct current resistance (DC-R) of IDE chips at 12 weeks post-implantation.

Chip no.	Resistance (DC-R, MΩ)
1	3.134
4	∞
6	∞
13	∞
18	1.568
20	∞
29	2.486
30	∞

**Table 5 micromachines-11-01064-t005:** Result of histopathological evaluation at 12 weeks post-implantation.

Sample	Test Sample			Controlled Sample		
Animal	#1	#2	#3	#1	#2	#3
Score						
Polymorphonuclear cells	0	4	1	2	0	0
Lymphocytes	4	5	4	4	4	4
Plasma cells	0	0	0	0	0	0
Macrophages	4	4	4	4	4	4
Giant cells	1	1	0	0	0	1
Necrosis	0	0	0	0	0	0
Subtotal (×2)	18	28	18	20	16	18
Neovascularization	0	0	0	0	0	0
Fibrosis	4	4	4	4	4	4
Fatty infiltrate	0	0	0	0	0	0
Subtotal	4	4	4	4	4	4
Total	22	32	22	24	20	22
Group total ^a^	22 + 32 + 22 = 76			24 + 20 + 22 = 66		
Average ^b^	Test-Control = 76/12 – 66/12 = 6.33 – 5.5 = 0.83					
Traumatic necrosis	0/12			0/12		
Foreign body debris	0/12			0/12		
No. sites examined	12			12		

^a^: Group total= Sum of score among groups; ^b^: Average = Sum of score among groups/the numbers of recognized implantation sites.

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
