# Peer review of "Bio-Compatibility and Bio-Insulation of Implantable Electrode Prosthesis Ameliorated by A-174 Silane Primed Parylene-C Deposited Embedment"

_micromachines, 2020, doi:10.3390/mi11121064_

Round 1

Reviewer 1 Report

See report enclosed.

Reviewer 2 Report

Reviewer’s comments

This manuscript describes the bio-insulation and biocompatibility of the parylene-C-embedded electrode. The authors evaluated the performance of parylene-C as an encapsulation material of the electrode by carrying out cytotoxicity, hemolysis, irritation and pathological implantation examinations according to ISO 10993 guidance. The study provides a thorough understanding for the in vivo compatibility of the parylene-C encapsulation and its potential for the further applications in implantable devices; therefore, I recommend acceptance of the manuscript after addressing the following points.

[1] A-174 silane coating was mentioned in the early part of the manuscript, but it was not described in the results and discussion sections. Please clearly describe if the silane was used for each experiment and how it affected the results.

[2] In Figure 2B, please provide a more detailed description of each image to compare the performance of each material as an encapsulation agent in saline.

[3] Please state what the reagent control is in the cytotoxicity test (Figure 3 and Table 1).

[4] Please include scale bars for the magnified images of Figure 2C and the histological sections of Figure 7, Figure S2, and Figure S3.

[5] Please show the histological sections of HDPE controlled samples for the comparison with the chip-implanted group.

[6] In page 2 line 2, please check if the cytotoxicity was quantitatively or qualitatively graded.

Round 2

Reviewer 1 Report

The authors have addressed my previous remarks.